# A MEKK1 – JNK mitogen activated kinase (MAPK) cascade module is active in *Echinococcus multilocularis* stem cells

**Kristin Stoll**, **Monika Bergmann**, **Markus Spiliotis**, **Klaus Brehm***

University of Würzburg, Institute of Hygiene and Microbiology, Würzburg, Germany

\* kbrehm@hygiene.uni-wuerzburg.de

**Data Availability Statement:** All relevant data are within the manuscript and its Supporting Information files.

## Abstract

### Background

The metacestode larval stage of the fox-tapeworm *Echinococcus multilocularis* causes alveolar echinococcosis by tumour-like growth within the liver of the intermediate host. Metacestode growth and development is stimulated by host-derived cytokines such as insulin, fibroblast growth factor, and epidermal growth factor via activation of cognate receptor tyrosine kinases expressed by the parasite. Little is known, however, concerning signal transmission to the parasite nucleus and cross-reaction with other parasite signalling systems.

### Methodology/Principal findings

Using bioinformatic approaches, cloning, and yeast two-hybrid analyses we identified a novel mitogen-activated kinase (MAPK) cascade module that consists of *E. multilocularis* orthologs of the tyrosine kinase receptor interactor *Growth factor receptor-bound 2*, EmGrb2, the MAPK kinase kinase EmMEKK1, a novel MAPK kinase, EmMKK3, and a close homolog to c-Jun N-terminal kinase (JNK), EmMPK3. Whole mount *in situ* hybridization analyses indicated that EmMEKK1 and EmMPK3 are both expressed in *E. multilocularis* germinative (stem) cells but also in differentiated or differentiating cells. Treatment with the known JNK inhibitor SP600125 led to a significantly reduced formation of metacestode vesicles from stem cells and to a specific reduction of proliferating stem cells in mature metacestode vesicles.

### Conclusions/Significance

We provide evidence for the expression of a MEKK1-JNK MAPK cascade module which, in mammals, is crucially involved in stress responses, cytoskeletal rearrangements, and apoptosis, in *E. multilocularis* stem cells. Inhibitor studies indicate an important role of JNK signalling in *E. multilocularis* stem cell survival and/or maintenance. Our data are relevant for molecular and cellular studies into crosstalk signalling mechanisms that govern *Echinococcus* stem cell function and introduce the JNK signalling cascade as a possible target of chemotherapeutics against echinococcosis.

**Funding:** This work was supported by the Wellcome Trust (https://wellcome.ac.uk/), grant 107475/Z/15/Z (FUGI), and by a grant of the Bayerische Forschungsstiftung (https://www.forschungsstiftung.de/) (AZ-1341-18) (both to KB). KS was supported by a grant of the Medical Faculty, University of Würzburg (GSLS/KS/270194; https://www.med.uni-wuerzburg.de/startseite/). The funders had no role in study design, data collection and analysis, decision to publish, or preparation of the manuscript.

**Competing interests:** The authors have declared that no competing interests exist.

## Author summary

The metacestode larva of the tapeworm *E. multilocularis* grows infiltrative, like a malignant tumour, within the liver of the host thus causing the lethal disease alveolar echinococcosis. Previous work established that the metacestode senses signals of host hormones and cytokines by expressing surface receptors that share high homology with respective host receptors. However, little is known how these signals are transmitted from the parasite cell surface to the nucleus to alter gene expression. In this work, the authors present a module of several protein kinases that typically transmit cytokine signals from surface receptors to central regulators called mitogen-activated protein kinases (MAPK). The authors demonstrate that this module is active in parasite stem cells, which drive the development of metacestode larva. They also show that inhibitors directed against one component of the module, EmMPK3, affect maintenance and/or survival of stem cells in the metacestode and prevent the formation of metacestode larva from parasite cell cultures. This information facilitates molecular and cellular studies to unravel the complex signalling network that regulate *Echinococcus* stem cell proliferation in response to host signals. Furthermore, these data could open new ways of anti-parasitic chemotherapy by introducing EmMPK3 as a possible drug target.

## Introduction

The metacestode larval stage of the fox-tapeworm *E. multilocularis* is the causative agent of alveolar echinococcosis (AE), a lethal zoonosis prevalent in the Northern Hemisphere [1]. During an infection of the intermediate host the parasite undergoes several developmental transitions that are driven by a population of pluripotent stem cells, called 'germinative cells' (GC) [2,3]. A small number of undifferentiated GC are delivered into the host as part of the oncosphere larva that is present in infectious eggs, which are released by the definitive host and are orally taken up by the intermediate host. Upon hatching in the intestine and penetration of the intestinal wall, the oncosphere gains access to the host liver where the GC drive the metamorphosis-like transition towards the metacestode stage, a meshwork of fluid-filled vesicles that grows infiltratively, like a malignant tumour, into the surrounding host tissue [3,4]. Parasite proliferation is exclusively driven by GC, which, as typical for flatworm stem cells, are the only mitotically active cells in the parasite and give rise to all differentiated cells (e.g. muscle, nerve, glycogen/lipid storing cells) [2]. Notably, the formation of the structurally unusual metacestode stage from the oncosphere is achieved by modification of the parasite's body axes in such a way that the anterior-posterior body axis of the oncosphere is given up and turned into the exclusively posteriorized tissue of the metacestode [5]. At the end of an infection in natural intermediate hosts (rodents), the anterior-posterior body axis is re-established when GC give rise to brood capsules which end up in the formation of protoscoleces [5], which are head-like structures that are transmitted to the definitive host when it takes the prey. In human infections, on the other hand, protoscoleces are rarely produced [6]. In general, human AE is very difficult to treat and chemotherapeutic treatment, directed against parasite β-tubulin, must be given for prolonged periods of time (sometimes life-long) and is associated with adverse side effects [7].

During recent years, we have established that host hormones and cytokines of the insulin- and the fibroblast growth factor (FGF) families stimulate metacestode development from GC as well as GC proliferation and growth of mature metacestode vesicles [8,9]. Furthermore, Cheng et al. [10] demonstrated that human epidermal growth factor (EGF) stimulates GC proliferation in mature metacestode vesicles. We had reported the presence of receptor tyrosine

kinases (RTK) of the insulin-, the FGF-, and the EGF-families in *E. multilocularis* and in all cases mentioned above, direct activation of the parasite RTKs through the cognate host hormone has been demonstrated [8–12]. Hence, whereas some information is available concerning the interaction of host cytokines with parasite receptors, little is currently known on how these signals are transmitted to the nucleus to alter gene expression patterns. In the case of RTKs, components of mitogen-activated protein kinase (MAPK) cascade modules are important downstream signalling factors. In these modules, MAPKs of the Erk-, p38-, and JNK-subfamilies are typically activated by upstream MAPK kinases (MAPKK), which are themselves regulated by upstream MAPKK kinases (MAPKKK), and MAPKKK kinases [13]. Once activated by upstream kinases, the Erk-, p38-, and JNK MAPKs eventually phosphorylate, and thereby activate, transcription factors such as c-Jun or c-Fos which then translocate to the nucleus and regulate specific gene expression [13]. In previous studies, we had characterized several of the respective compounds in *E. multilocularis* of which three, the Erk-like MAPK EmMPK1, the MAPKK EmMKK2, and the MAPKKK EmRaf formed one MAPK cascade module as assessed by yeast two-hybrid analyses [14–16]. We also demonstrated that the *E. multilocularis* p38-like MAPK EmMPK2 is a constitutively active kinase that acts independently of upstream activation by MAPKKs [17]. Concerning the third branch of MAPKs, the c-Jun activating JNKs, no studies have so far been conducted in *E. multilocularis*.

Classically, the JNK pathway had been studied in the context of stress responses but is meanwhile also known as being involved in diverse processes such as cytoskeletal rearrangements or apoptosis [18]. JNKs are activated by diverse upstream receptors such as members of the GPCR family (G-protein coupled receptors), the TNF-α receptor, and several RTKs [18]. One of the best studied pathways leading to JNK activation involves direct binding of the MAPKKK MEKK1 to JNK after stimulation of upstream EGF receptors, using Grb2 (*Growth factor receptor bound 2*) as an adapter molecule between the EGF receptor and MEKK1 [18–22]. Interestingly, in free-living planarians, which are closely related but non-parasitic relatives to cestodes, both orthologs to MEKK1 and JNK are involved in processes of body axis formation and stem cell (neoblast) maintenance [23–26]. Given that the *E. multilocularis* metacestode is a result of modified body axis formation [5] and that neoblast-like stem cells are central to the proliferation of *Echinococcus* larvae [2], this prompted us to closer characterize this specific branch of MAPK signalling in parasite larvae. We herein report that close orthologs of MEKK1 and JNK are both expressed in *E. multilocularis* stem cells and functionally interact in a manner similar to their mammalian counterparts. We also report anti-parasitic effects of a known and specific inhibitor of JNK, SP600125.

## Methods

### Ethics statement

*In vivo* propagation of parasite material was performed in mongolian jirds (*Meriones unguiculatus*), which were raised and housed at the local animal facility of the Institute of Hygiene and Microbiology, University of Würzburg. This study was performed in strict accordance with German (*Deutsches Tierschutzgesetz,TierSchG*, version from Dec-9-2010) and European (European directive 2010/63/EU) regulations on the protection of animals. The protocol was approved by the Ethics Committee of the Government of Lower Franconia (Regierung von Unterfranken) under permit number 55.2–2531.01-61/13.

### Organisms and culture methods

All experiments were performed with the *E. multilocularis* isolates H95, GH09, Ingrid, and J2012 [27] which either derive from a naturally infected fox of the region of the Swabian

Mountains, Germany (H95) [28] or from Old World Monkey species (*Macaca fascicularis*) that had been naturally infected in a breeding enclosure (GH09, Ingrid, J2012) [29]. The isolates were continuously passaged in mongolian jirds (*Meriones unguiculatus*) essentially as previously described [30]. During long-time peritoneal passage, *E. multilocularis* metacestode tissue gradually loses the capacity to form brood capsules and protoscoleces [30] and at the time of these experiments, isolate J2012 (from 2012) was still producing protoscoleces, albeit significantly reduced when compared to fresh isolates, whereas H95 (1985), GH09, and Ingrid (both from 2009) no longer developed brood capsules neither *in vivo* nor *in vitro*. *In vitro* culture of parasite metacestode vesicles under axenic conditions was performed as previously described [30,31] and the isolation and maintenance of primary parasite cell cultures was carried out essentially as established by Spiliotis et al. [32,33]. In all cases, media were changed every three days (d). For inhibitor studies, specific concentrations of SP600125 (Selleckchem, München, Germany), dissolved as 100 mM stock solution and stored at -80˚C, were added to parasite cultures as indicated and as negative control DMSO (0.1%) was used.

### Nucleic acid isolation, cloning and sequencing

RNA isolation from *in vitro* cultivated axenic metacestode vesicles and primary cells was performed using a Trizol (5Prime, Hamburg, Germany)-based method as previously described [8]. For reverse transcription, 2 µg total RNA was used for cDNA synthesis using oligonucleotide CD3-RT (5'-ATC TCT TGA AAG GAT CCT GCA GGT$_{26}$ V-3'). PCR products were cloned using the PCR cloning Kit (QIAGEN, Hilden, Germany) or the TOPO XL cloning Kit (Invitrogen). The complete list of primer sequences used for *emmekk1*, *emgrb2*, *emmkk3*, *emmkk4*, *emmkk5*, and *emmpk3* cDNA amplification and characterization is given in S1 Table. Identification of genes of interest is discussed in conjunction with results below. Upon cloning, PCR products were directly sequenced using primers binding to vector sequences adjacent to the multiple cloning site by Sanger Sequencing (Microsynth Seqlab, Göttingen, Germany). The sequences of all genes newly characterized in this work have been submitted to the GenBank, EMBL, and DDJB databases under accession numbers listed in S1 Table.

### *In situ* hybridization and 5-ethynyl-2'-deoxyuridine (EdU) labeling

Digoxygenin (DIG)-labeled probes were synthesized by *in vitro* transcription with T7 and SP6 polymerase (New England Biolabs), using the DIG RNA labelling kit (Roche) according to the manufacturer's instructions from *emmekk1-* and *emmpk3*-cDNA fragments cloned into vector pJET1.2 (Thermo Fisher Scientific). Primers for probe production are listed in S1 Table. The probes were subsequently purified using the RNEasy Mini Kit (Qiagen), analysed by electrophoresis, and quantified by dot blot serial dilutions with DIG-labeled control RNA (Roche). Whole-mount *in situ* hybridization (WISH) was subsequently carried out on *in vitro* cultivated metacestode vesicles essentially as previously described [2,5], using vesicles of at least 1 cm in diameter to avoid losing material during washing steps. Fluorescent specimen were imaged using a Nikon A1 confocal microscope and maximum projections created using ImageJ as previously described [5]. In all cases, negative control sense probes yielded no staining results. *In vitro* labelling with 50 µM EdU was done for 5 hours (h) and fluorescent detection with Alexa Fluor 555 azide was performed after WMISH essentially as previously described [2].

### Yeast Two hybrid (Y2H) analyses

The Gal4-based Matchmaker System (Clontech) was used as described by Zavala-Góngora et al. [34]. The coding sequences of *E. multilocularis* genes were amplified by PCR from a pool of cDNA obtained by reverse transcription of RNA from metacestode vesicles, protoscoleces,

and primary cells, as previously described [5]. In most cases, the complete coding sequence was amplified, except in the case of *emmekk1* for which regions encoding N-terminal and C-terminal parts of the protein had to be cloned separately. The primers used for amplification of all cDNAs are provided in S1 Table and included adapter sites for the appropriate restriction enzymes for cloning into plasmids pGADT7 (for fusions to the GAL4 activation domain, AD) and pGBKT7 (for fusions to the GAL4 DNA binding domain, BD) as indicated. For several genes analysed in this work, Y2H fusion vectors had previously been generated by us and are described in [14] (*emraf*), [35] (*axin1*, *axin2*), and [16] (*emmekk1*, *emmkk2*).

pGADT7- and pGBKT7-fusion plasmids were co-transformed into yeast strain AH109 and grown on leucine/tryptophane-deficient minimal medium plates. After three days of incubation, positive transformants were selected (three independent clones for each combination) and interaction analysis was performed for another three days at 30˚C on minimal medium plates lacking leucine, tryptophane (growth control), histidine (medium stringency), or histidine and adenine (high stringency). Co-transformants of T-antigen-pGADT7 with p53-pGBKT7 served as positive control, those with T-antigen-pGADT7 and lamC-pGBKT7 as negative controls. In cases of autoactivation, indicated by colony growth after co-transformation of fusion proteins with empty vector controls, interaction studies were additionally performed on minimal medium lacking leucine, tryptophane and histidine with 1-25mM 3-Amino-1,2,4- triazol (3-AT) at 30˚C for one week.

### RT-qPCR on hydroxy urea (HU)- and Bi2536 treated metacestode vesicles

Metacestode vesicles of isolate Ingrid were *in vitro* cultivated under axenic conditions for 7 d essentially as previously described [2] and were then treated with either 40 mM HU [2] or 100 nM of the Polo-like kinase inhibitor Bi 2536 [36] for another 7 d to specifically eliminate the germinative cell population (medium change and inhibitor addition every 2 d). Total RNA was isolated from inhibitor-treated vesicles and control vesicles (incubated with DMSO) and subjected to cDNA synthesis as previously described [37]. qPCR was performed in a StepOne Plus cycler (Applied Biosystems, U.S.A.) using a PCR mix of 5 x HOT FIRE-Pol EvaGreen qPCR Mix plus 2.4 µl (1x), 0.72 µl (300 nM) of each primer, 6.96 µl of DNase free water and cDNA. Cycling conditions were: 15 min at 95˚C, followed by 40x (15 s at 95˚C, 20 s at 60˚C, 20 s at 72˚C). Data collection was performed at 72˚C. Normalization was carried out using the constitutively expressed gene *elp* (EmuJ_000485800) as previously described [37]. Primer sequences, amplicon size and PCR efficiency values for primers are described in S1 Table.

### Inhibitor studies

Inhibitor studies were carried out on mature metacestode vesicles and primary cell cultures according to previously established protocols [8,9,17,36]. Mature metacestode cultures were established as described in [30] until they reached a vesicle size of about 5 mm when inhibitor SP600125 (Selleckchem) was added at concentrations between 1 and 25 µM (stock solution 100 mM in DMSO). Medium was changed and new inhibitor was added every three d for up to 13 d of incubation. In negative controls, only DMSO was added. Structural integrity of the vesicles was monitored daily by light microscopy. At day 13, new medium without inhibitor was added and the vesicles were left to recover for 3 h before a 5 h EdU incorporation pulse was carried out essentially as previously described [36]. Primary cell cultures were prepared as previously described [30] with media changes and addition of inhibitor SP600125 every three d for up to 19 d. The formation of fully mature metacestode vesicles was monitored light microscopically essentially as described previously [9,36]. All experiments were carried out with three biological triplicates and three technical triplicates per metacestode or primary cell culture.

## Computer analyses and statistics

Amino acid comparisons were performed using BLASTP on the nr-aa and SWISSPROT database collections available under (https://www.genome.jp/). Genomic analyses and BLAST searches against the *E. multilocularis* genome [27] were done using resources at (https://parasite.wormbase.org/index.html). CLUSTAL W alignments were generated using MegAlign software (DNASTAR Version 12.0.0) applying the BLOSUM62 matrix. Domain predictions were carried out using the simple modular architecture research tool (SMART) available under (http://smart.embl-heidelberg.de/) as well as PROSITE scans available under (https://prosite.expasy.org/scanprosite/). Two-tailed, unpaired student's T-tests were performed for statistical analyses (GraphPad Prism, version 4). Error bars represent standard error of the mean. Differences were considered significant for p-values below 0.05 (indicated by *).

## Results

### Cloning and characterization of an *E. multilocularis mekk1* ortholog

Due to its central role in MAPK cascade signaling in mammals and its function in body axis determination in planarians we were interested in identifying and cloning a *mekk1* ortholog of *E. multilocularis*. To this end, we first mined the available *E. multilocularis* genome information [27] by BLASTP analyses using human MEKK1 as a query and yielded as best hit a protein encoded by locus EmuJ_000389600 (annotated as a MAPKKK). Next, we used in similar BLASTP analyses a previously described MEKK1 protein of planaria [26] and, again, obtained EmuJ_000389600 as the gene encoding a protein with highest similarities. Informed by the *E. multilocularis* genome sequence we then designed primers to clone and sequence the full-length EmuJ_000389600 cDNA from metacestode mRNA preparations, thus verifying the annotated version as depicted in WormBase Parasite.

The EmuJ_000389600 locus comprises 13 exons and 12 introns, of which two are 5' introns that are located between transcription start site and the translational start codon. The full-length transcript comprises 5.047 kb with a 5' UTR of 281 bp and a 3' UTR of 377 bp. The open reading frame comprises 4.386 kb (excluding the stop codon) and encodes a protein of 1,462 amino acids. Hallmarks of MEKK1 factors are a protein kinase domain and a so-called RING motif which resembles a Zink finger that, at least in mammals, acts as ubiquitin ligase that mediates degradation of target proteins [38]. Accordingly, by *in silico* protein domain analyses we identified a protein kinase domain between residues 1096 and 1363 as well as a putative RING motif between residues 339 and 402 of the EmuJ_000389600 product (Fig 1). Finally, when we used the EmuJ_000389600 deduced amino acid sequence as a query in reciprocal BLASP searches against the SWISSPROT database, we found highest homologies with mammalian MEKK1 orthologs (e.g. 52% identical and 72% similar residues to human MEKK1 in the kinase domain). Similarly, highest homologies were found in reciprocal BLASTP analyses to planarian MEKK1 (Fig 1). Taken together, based on the domain structure of the protein encoded by EmuJ_000389600 which contained all hallmarks of MEKK1 kinases and the homologies of this protein to human and planarian MEKK1 orthologs, we concluded that we had successfully cloned the *E. multilocularis mekk1* locus. We thus named the gene *emmekk1* (for *E. multilocularis* MEKK1) and the respective protein EmMEKK1.

### Expression of *emmekk1* in *E. multilocularis* stem cells

According to transcriptome analyses that had been produced during the *E. multilocularis* whole genome project [27], *emmekk1* displayed higher expression in primary cell cultures after 2 or 7 d of incubation when compared to metacestode vesicles (S1 Fig). Since these

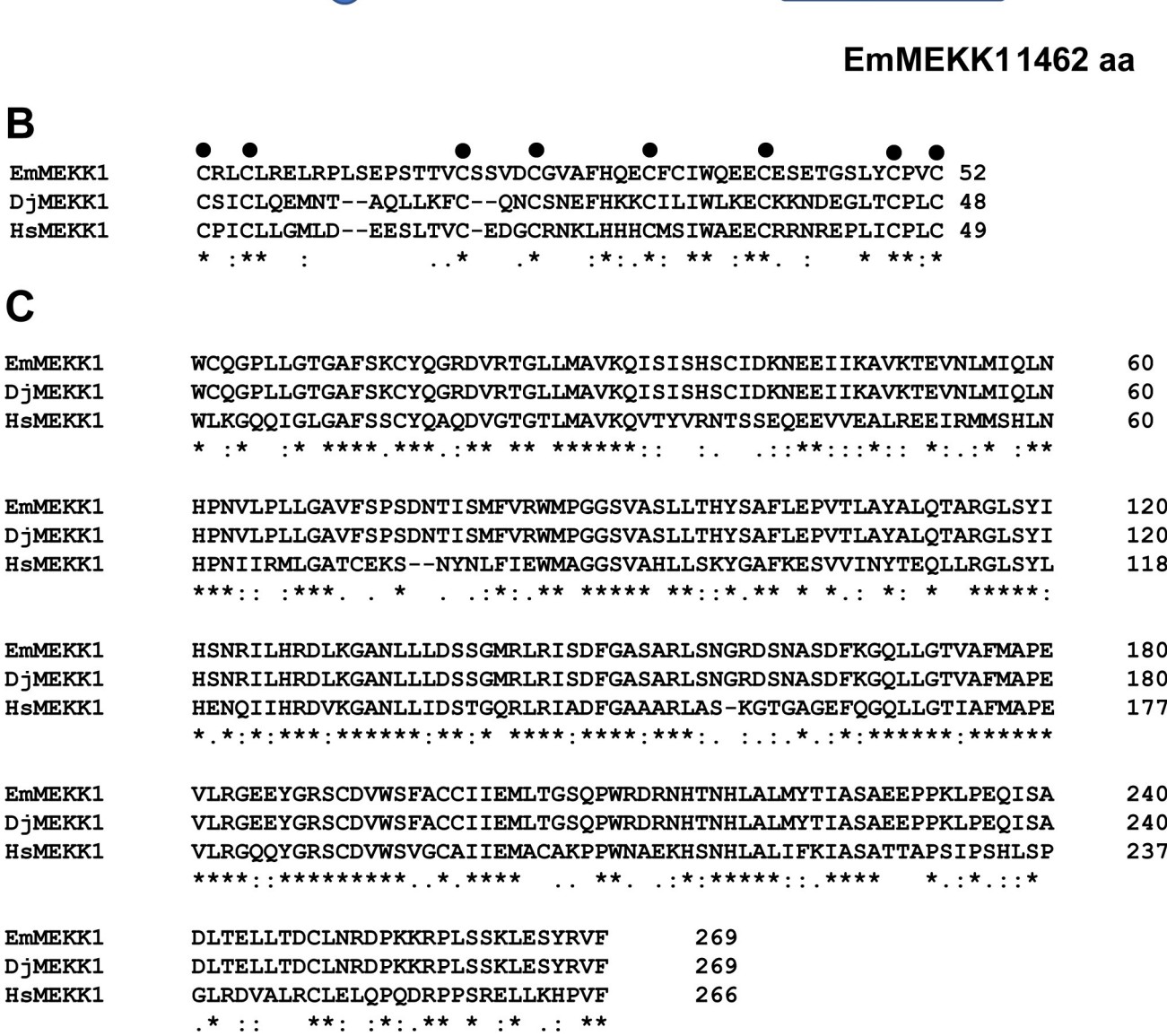

**Fig 1. Sequence features of EmMEKK1.** (A) EmMEKK1 domain structure. Shown in blue are the locations of the RING domain (R) and the serine/threonine kinase domain (S/TKD). (B) Amino acid sequence alignment of the RING domains of different MEKK1 orthologs of *E. multilocularis* (EmMEKK1; this work), *Dugesia japonica* (DjMEKK1; DDBJ/EMBL/GenBank accession no.: BBA10910) and *H. sapiens* (HsMEKK1; Q13233). Sites of perfect alignment (*) as well as groups of strong (;) or weak (.) similarity are marked below the alignment. Highly conserved cysteine residues of the RING zink-finger are additionally marked by a dot above the alignment. (C) Amino acid sequence alignment of serine/threonine kinase domains of different MEKK1 orthologs. Sequence origins and decorations are as in (B).

cultures are highly enriched in GC [2], we assumed that *emmekk1* might show a dominant expression in this cell type. To further investigate this aspect, we performed RT-qPCR analyses on metacestode vesicles in which the GC population had been depleted after 7 d treatment with the inhibitors HU and Bi 2536. As we have previously demonstrated, both inhibitors specifically eliminated stem cells in metacestode vesicles which otherwise remained structurally

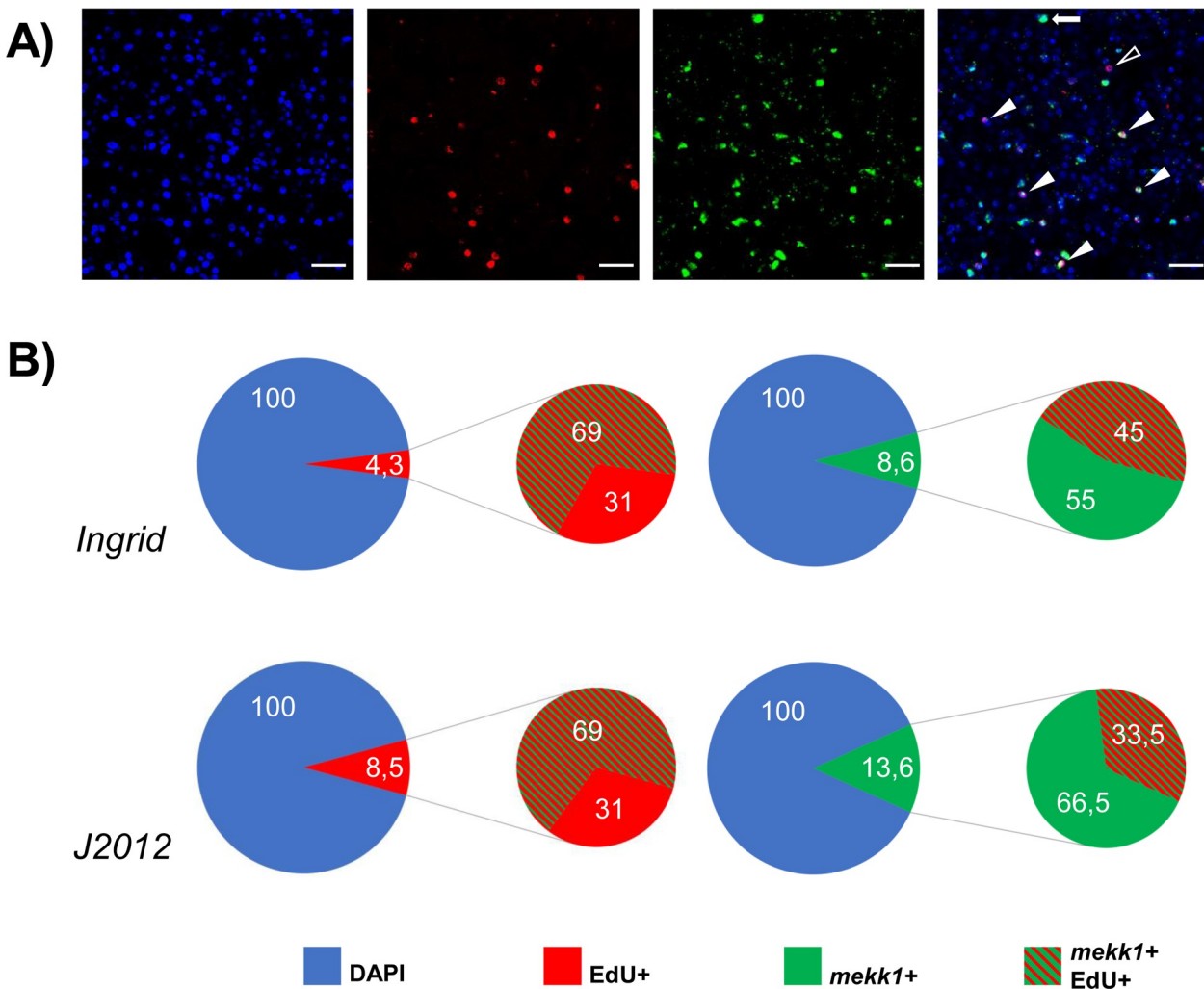

**Fig 2. Expression of *emmekk1* in metacestode vesicles.** A) WISH on vesicle germinal layer. Shown are from left to right: nuclear DAPI staining (blue), EdU detection (red), *emmekk1* WISH (green), and merge of all channels. Full triangles indicate cells with double staining for EdU and the *emmekk1*-probe, open triangle indicates EdU+ alone, arrow indicates *emmekk1*+ alone. Bar represents 20 µm. B) Summary of the WISH results. Values are given in percent for isolates Ingrid (upper panel) and J2012 (lower panel). The color code is indicated below.

intact [2,36]. As shown in S3 Fig, compared to the constitutively expressed control gene *elp* [37], the expression of *emmekk1* was reduced to ~ 40% in HU treated vesicles and to ~60% in Bi 2536 treated vesicles, further indicating expression of the gene in the GC population. To clarify the situation, we carried out WISH analyses on metacestode vesicles that had been incubated with EdU, thus identifying the proliferating stem cell compartment [2]. As shown in Fig 2, in metacestode vesicles of two different *E. multilocularis* isolates we detected *emmekk1* signals in both EdU+ and EdU- cells. During the 5 h EdU pulse, metacestode vesicles of the two *E. multilocularis* isolates Ingrid and J2012 showed slightly different percentages of EdU+ cells (4.3 +/- 2.4% in Ingrid, n = 1712; 8.5 +/- 2.9% in J2012, n = 2019) and of *emmekk1*+ cells (8.6 +/- 3.6% in Ingrid, n-1712; 13.6 +/- 4.8% in J2012, n = 2019). In both isolates, on the other hand, co-localization values for EdU+/*emmekk1*+ of all EdU+ cells were around 69% (69.2 +/- 14,2% in Ingrid, n = 162; 68.8 +/- 23.2% in J2012, n = 70). With reference to all *emmekk1* + cells, co-localization values were slightly higher for J2012 (45.0 +/- 16.0%, n = 269) than for

Ingrid (33.5 +/- 13.8%, n = 142). Hence, whereas in both isolates roughly two thirds of all stem cells that underwent DNA synthesis also expressed *emmekk1*, the ratio of *emmekk1+* cells in S-phase was slightly higher in J2012 than in Ingrid (although not statistically significant). Differences in the number of EdU+ cells in metacestode vesicles of distinct isolates probably resulted from differences in vesicle size, due to slow equilibration between EdU in the medium and the large amount of hydatid fluid within the vesicles [2]. Since isolate Ingrid routinely produces larger vesicles than J2012 within the same time of *in vitro* cultivation, this could account for the slight differences in the number of EdU+ cells observed. Nevertheless, in both cases we detected comparable numbers of EdU+ cells that stained positive for *emmekk1*, showing that the expression of *emmekk1* in GC is not a peculiar trait of only one isolate. Taken together, these analyses indicated that *emmekk1* is expressed in the majority of *E. multilocularis* stem cells, but also in a certain number of post-mitotic cells.

## Upstream interaction partners of EmMEKK1

Next, we were interested in identifying intracellular interaction partners of EmMEKK1 and, as in several previous studies on *E. multilocularis* signal transduction systems [14,16,34,35,39,40], employed the Y2H system for measuring protein-protein interactions. Since several attempts to full-length clone the *emmekk1* cDNA into AD- and BD- vectors failed, we decided to clone the reading frame as two parts. Plasmids pGADT7-5'mekk1 and pGBDT7-5'mekk1 encoded fusion proteins of the GAL4 AD or BD, respectively, with amino acid 5 to 803 of EmMEKK1 (pGADT7-5'mekk1) or amino acid 5 to 790 of EmMEKK1 (pGBDT7-5'mekk1). These plasmids thus contained the coding region for the RING domain but excluded the TKD. Through plasmids pGADT7-3'mekk1 and pGBDT7-3'mekk1, fusion proteins of AD and BD with coding regions for amino acid 729 to the C-terminus of EmMEKK1 were expressed. They thus contained the coding information for the TKD but excluded the RING domain.

As one of the most important interaction partners of mammalian MEKK1, growth factor receptor-bound protein 2 (Grb2) has previously been identified and shown to mediate the link to activated EGF receptors [21]. Since no gene encoding a Grb2 ortholog has so far been described in cestodes, we used the human Grb2 amino acid sequence and mined the available *E. multilocularis* genome information by BLASTP and reciprocal BLASTP searches. These analyses revealed that the *E. multilocularis* genome contains only one respective gene (EmuJ_000587600), which encodes a protein of the expected domain structure (SH3-SH2-SH3) with clear homologies (52% identical, 72% similar residues) to human Grb2. We thus named this gene *emgrb2*, encoding the protein EmGrb2 (Fig 3). Transcriptome analyses [27] indicated that *emgrb2* is expressed throughout the entire life cycle (S1 Fig).

Among the known interaction partners of mammalian MEKK1 are also the cancer-associated MAPKKK c-Raf [40] and axins, which are involved in the *wnt* signalling pathway [41,42]. Respective orthologs have already been characterized by us [14,35] and were thus included in the analyses as plasmids pGADT7-emraf/pGBDT7-emraf (EmRaf, EmuJ_001079900), pGADT-7-ax1/pGBDT7-ax1 (axin 1, EmuJ_000624800), and pGADT7-ax2/pGBDT7-ax2 (axin 2, EmuJ_001141200).

As shown in Fig 3, we could not detect interactions between EmMEKK1 and EmRaf as well as between EmMEKK1 or any of the two *Echinococcus* axins, at least in the Y2H system. For EmMEKK1 and EmGrb2, on the other hand, interactions were observed in the combinations EmGrb2-AD x 5'-MEKK1-BD as well as EmGrb2-AD x 3'-MEKK1-BD. Unfortunately, we already observed yeast growth under low stringency conditions when EmGrb2 was fused to the Gal4 BD and tested against the empty AD vector, so that only combinations with EmGrb2 fused to the AD could be analyzed. Nevertheless, since the interactions were observed under

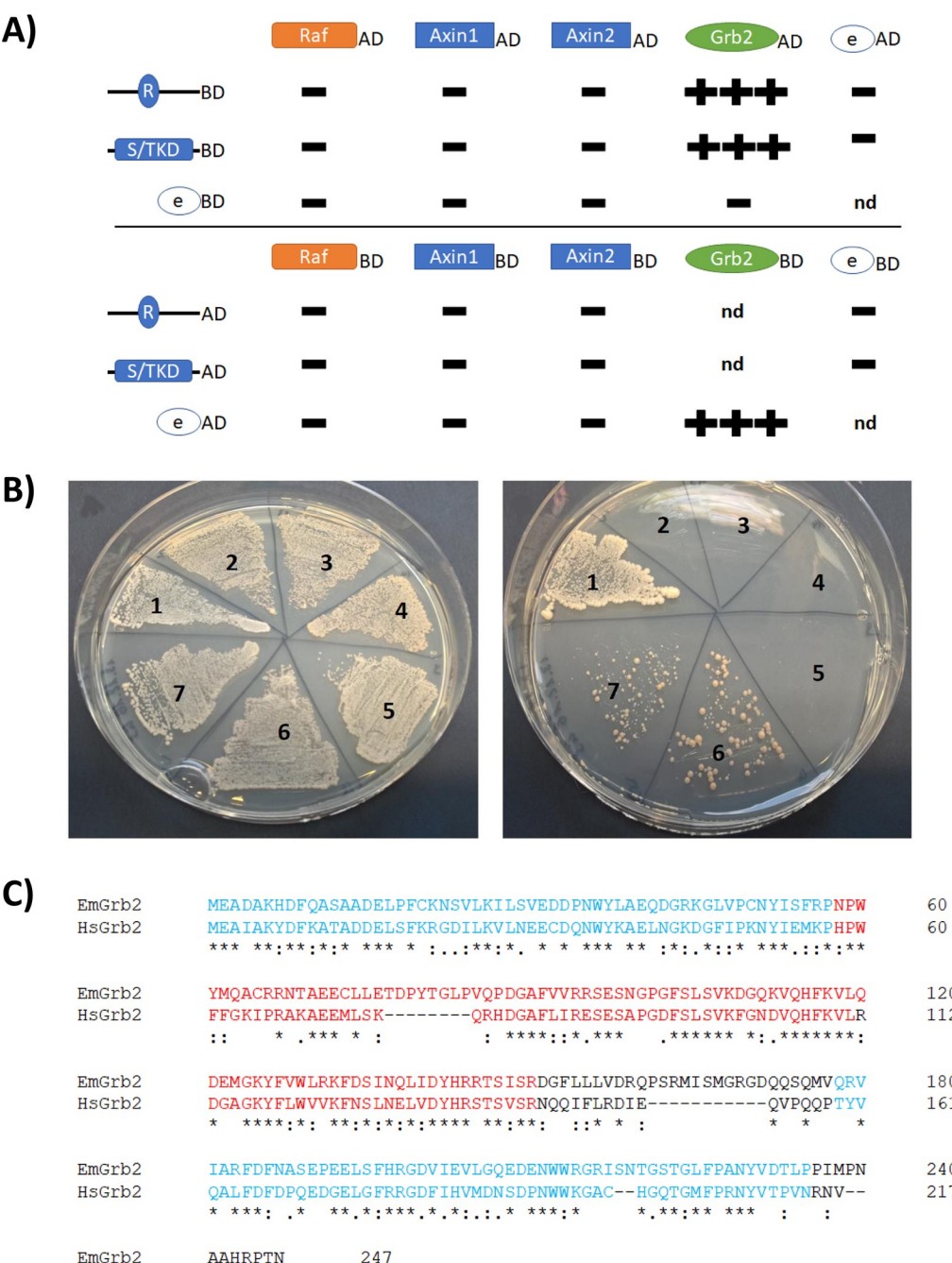

**Fig 3. Upstream interaction partners of EmMEKK1.** A) Summary of Y2H interaction experiments with putative upstream interaction partners. N-terminal and C-terminal regions of EmMEKK1 are indicated to the left as regions encompassing the RING- and the serine/threonine kinase domain, respectively, fused to Gal4-AD or Gal4-BD as indicated. Possible interaction partners tested in this study are indicated above as AD- and BD-fusions. Empty vector controls are marked by "e". No interaction (-) or interactions under high stringency conditions (+++) are indicated. nd = not determined. B) Y2H experiment showing the interaction between EmMEKK1 and EmGrb2. Left plate, SD-Leu-Trp for transfection control; right plate, SD-Leu-Trp-His-Ade for high stringency interaction. 1, positive control (T-antigen-AD x p53-BD); 2, negative control (T-antigen-AD x lamC-BD); 3, MEKK1-3'-AD x empty-BD; 4, MEKK1-5'-AD x empty-BD; 5, EmBrb2-AD x empty BD; 6, EmGrb2-AD x MEKK1-3'-BD; 7, EmGrb2-AD x MEKK1-5'-BD. C) Amino acid sequence alignment between EmGrb2 (above) and human Grb2 (HsGrb2). Identical residues are marked by "*" and biochemically similar residues by ":". SH3-domains are indicated in blue, the SH2 domain in red.

high stringency conditions (15 mM 3-AT), these analyses indicated that EmMEKK1 acts as a signal transducer downstream of EmGrb2.

## Downstream interaction partners of EmMEKK1

Typical downstream interaction partners of MAPKKKs are dual-specific MAPKKs, of which we had previously described two in *E. multilocularis* [16]. While both these MAPKKs, EmMKK1 and EmMKK2, had interacted with EmRaf as an upstream partner in Y2H analyses, only EmMKK2 interacted with the Erk-like MAPK EmMPK1 [16]. To identify the full set of MAPKKs that are encoded in the *E. multilocularis* genome, we, again, carried out BLASTP and reciprocal BLASTP searches using the full set of human MAPKKs as a query. By these analyses we identified three additional genes that encoded dual-specific MAPKKs and designated the genes and proteins according to the established nomenclature *emmkk3*/EmMKK3 (EmuJ_000123600), *emmkk4*/EmMKK4 (EmuJ_000221400), and *emmkk5*/EmMKK5 (EmuJ_001114500). As shown in Fig 4, phylogenetic analyses indicated close homologies between EmMKK3 and EmMKK5 to human MKK7 and MKK4 (both JNK branch), of EmMKK2 to human MEK1/MEK2 (Erk1/2 branch), and somewhat distant relatedness of EmMKK4 and EmMKK1 to human MKKs of the Erk- branch. No *Echinococcus* MAPKK showed close homologies to human orthologs of the p38-branch. According to available transcriptome data [27], all five MKK encoding genes were expressed in *E. multilocularis* larval stages with somewhat higher expression levels of *emmekk1*, *emmekk4*, and *emmekk5* in the protoscolex when compared to the metacestode (S1 Fig).

To identify downstream interaction partners for EmMEKK1, the full-length reading frames of all five *E. multilocularis* MKKs were cloned into pGADT7 and pGBKT7 and tested against the respective EmMEKK1 clones in the Y2H system. Since several of the EmMKKs already induced yeast growth when fused to the Gal4 AD in combination with the empty Gal4 BD vector, we subsequently only analyzed combinations of EmMEKK1 (5' and 3') fused to the AD versus the EmMKKs fused to the Gal 4 BD and applied medium to high stringency conditions. In these experiments we did not observe interactions of the EmMEKK1 3' part with any of the EmMKKs. For the combination of EmMEKK1 5' with EmMKK3, on the other hand, medium stringency interaction was consistently observed (Fig 4). Hence, like in mammals where MAPKKs of the JNK branch (MKK4 and MKK7) are interacting with MEKK1 [43], *E. multilocularis* also appears to employ at least one interacting pair of a MAPKK of the JNK branch (EmMKK3) an a MEKK1 ortholog (EmMEKK1).

## The *E. multilocularis* JNK EmMPK3 as downstream interaction partner of EmMKK3

Of the three major branches of MAPKs (Erk, p38, JNK), we had previously characterized the Erk-like EmMPK1 [15] and the p38-like EmMPK2 [17] in *E. multilocularis*, but so far no member of the JNK family. To investigate whether *Echinococcus* also expresses the third branch of MAPKs, we thus carried out BLASTP and reciprocal BLASTP analyses on the *E. multilocularis* genome using human JNK1 as a query. Highest homologies of 61% identical amino acid residues and 76% similar residues were found for a protein encoded by locus EmuJ_000174000 which, when BLAST analyzed against the SWISSPROT database, also revealed highest homologies to all three human JNK isoforms as well as to JNKs from other organisms. MAPKs can be distinguished by a hallmark TXY motif within the kinase domain activation loop which reads TEY in the case of Erk-like kinases, TGY for p38 MAPKs, and TPY for JNK [44]. Accordingly, EmuJ_000174000 encoded a kinase with TPY in the activation loop that overall displayed clear homologies to human JNK1 and planarian JNK (Fig 5). We

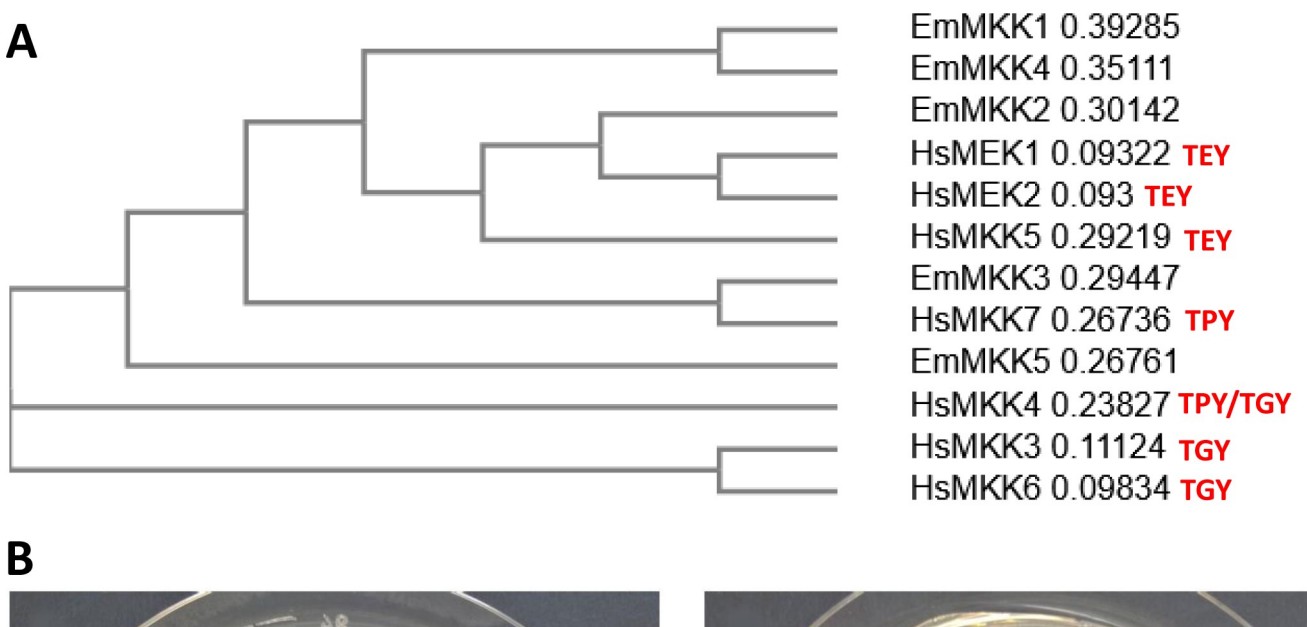

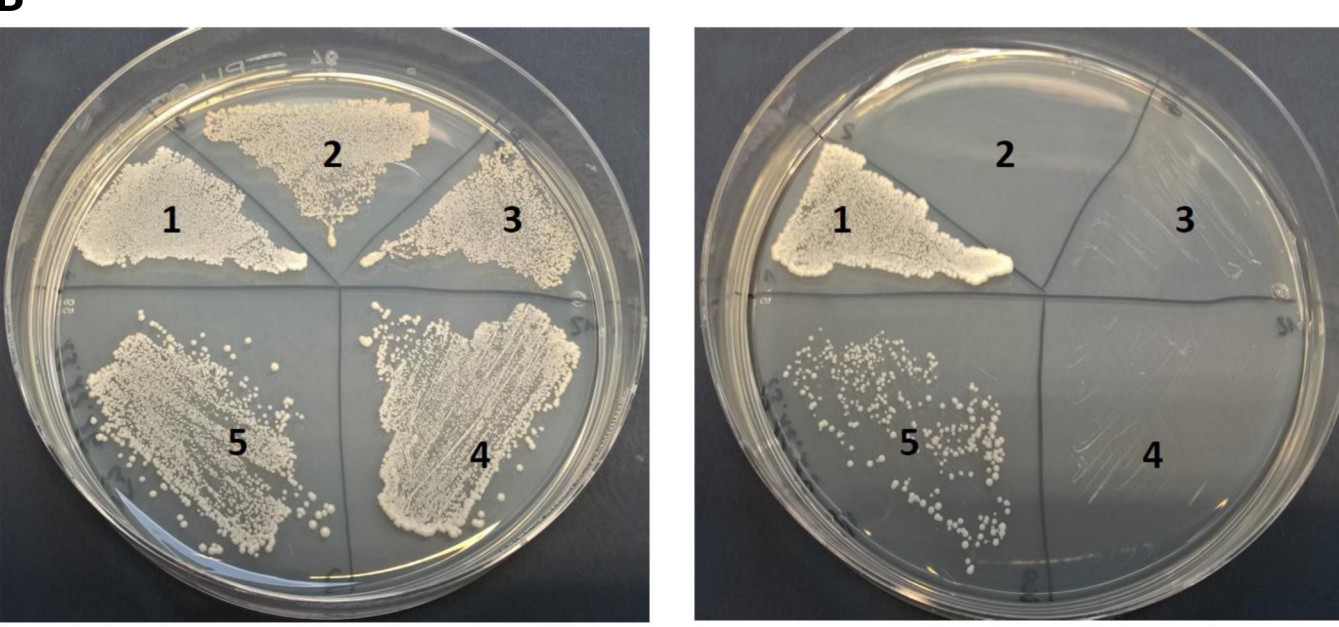

**Fig 4. Interaction between EmMEKK1 and EmMKK3.** A) Phylogenetic tree of *Echinococcus* MAPKK (as characterized in this work) and human MAPKK. CLUSTALW alignment was performed with full-length amino acid sequences of *E. multilocularis* EmMKK1 (accession no. FN434110), EmMKK2 (FN434111), EmMKK3 (MW358635), EmMKK4 (MW358636), and EmMKK5 (MW358637) as well as human (Hs) MKK1 (Q02750), MKK2 (P36507), MKK3 (P46734), MKK4 (P45985), MKK5 (Q13163), MKK6 (P52564), and MKK7 (O14733). Node distances are indicated to the right. Specificities of human enzymes to MAPK of the Erk-branch (TEY), the JNK-branch (TPY) and the p38-branch (TGY) are indicated in red. B) Y2H experiment showing the interaction between EmMEKK1 and EmMKK3. Left plate, SD-Leu-Trp for transfection control; right plate, SD-Leu-Trp-His with 7,5 mM 3-Amino-1,2,4-triazol. 1, positive control (T-antigen-AD x p53-BD); 2, negative control (T-antigen-AD x lamC-BD); 3, EmMEKK1-5'-AD x empty-BD; 4, EmMKK3-BD x empty-AD; 5, EmMEKK1-5'-AD x EmMKK3-BD.

thus named the gene *emmpk3*, which encodes the JNK-like *E. multilocularis* protein EmMPK3. Transcriptome data [27] indicated that *emmpk3* is well expressed in metacestode vesicles and protoscoleces.

To investigate whether EmMPK3 interacts with the MKK4/7-like MAPKK EmMKK3 we, again, used the Y2H system, cloned the full-length reading frame of EmMPK3 into pGADT7, and tested it against the EmMKK3 BD fusions mentioned above. As shown in Fig 5, EmMPK3

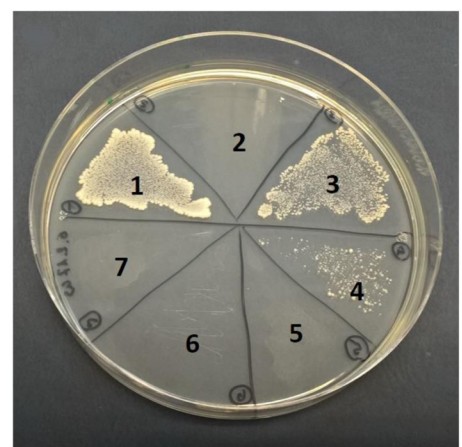

**A**

**B**

```
                                                        +             +
HsJNK1    ----------MSRSKRDNNFYSVEIGDSTFTVLKRYQNLKPIGSGAQGIVCA------- 42
EmMPK3    MPVTETSTEYSPSIPQIPPNFHIETVGESQFVIPDRYSGLRPIGTGAQGYVVS------- 53
SmedJNK   ----------MAQGSFNERFHTEVIGDSRFTILNRYSNLRAIGSGAQGYVVILPISYRA 49
                    :   .    .*:   :*:* *.: .**..*: **:**** *
                    +                              +
HsJNK1    AYDAILERNVAIKKLSRPFQNQTHAKRAYRELVLMKCVNHKNIIGLLNVFTPQKSLEEFQ 102
EmMPK3    AFDSVSQRKVAIKKLTHPFQNVTHAKRAYREIVLMKLVNHRNIISLLDAFSPQSTLDEFR 113
SmedJNK   ALDSVTGQQVAIKKLARPFQNVTHAKRAYREFILMKLVNHKNIIGLLNAFTPQQTINNFQ 109
          * *::  ::******::**** *********::*** ***:***.**:.*:**.:::*:
              + +                                +
HsJNK1    DVYIVMELMDANLCQVIQMELDHERMSYLLYQMLCGIKHLHSAGIIHRDLKPSNIVVKSD 162
EmMPK3    DVYLVMELMDASLSHVIHMELDHERFSFLLYQILCGLKHLHVAGIIHRDLKPNNIAVKHD 173
SmedJNK   DLYLVMELMDANLCQVINMDLDHERTSYLLYQLLCGVKHLHAAGIIHRDLKPSNIVVKHD 169
          *:*:*******.*.:**:*:***** *:****:***:**** **********.**.** *
              +                    ___
HsJNK1    CTLKILDFGLARTAGTSFMMTPYVVTRYYRAPEVILGMGYKENVDLWSVGCIMGEMVCHK 222
EmMPK3    CTLKILDFGLARSGTENFMMTPYVVTRYYRAPEVILGMGYTANVDIWSVGCIFAEMVLER 233
SmedJNK   CTLKILDFGLARSAGDSFLMTPYVVTRYYRAPEVILGMGYTENVDIWSVGCIFAEMVCER 229
          *************:.   .*:*****************.  ***:******:.*** .:

HsJNK1    ILFPGRDYIDQWNKVIEQLGTPCPEFMKKLQPTVRTYVENRPKYAGYSFEKLFPDVLFPA 282
EmMPK3    TFFPGTDHIDQWSKITAELGTPSRDFIDRLDRDVRNYVLSRPIVPRRSFETLFPDDVFPE 293
SmedJNK   IMFPGTDHIDQWTKITQLLGTPSDEFLSRLQPSVRNYVQSRPRTFGKSFDDLFPDDCFPE 289
           :*** *:****.*:    ****. :*::*: **.** .** :  **: ****  **

HsJNK1    DS-EHNKLKASQARDLLSKMLVIDASKRISVDEALQHPYINVWYDPSEAEAPPPKIPDKQ 341
EmMPK3    PSRRHADLNPAMARDLLSRMLVIDPVERITVGEALRHPYVSLWSDDAEINGPPPGCYDPD 353
SmedJNK   PSLEHATLNAYWARDLLKKMLVIDPLQRISVGEALRHQYINVWFEDYEVNGPPPGQYDHS 349
              * .* *:   *****.:***** :**:.***:* *:..* :   * :.***   * .

HsJNK1    LDEREHTIEEWKELIYKEVMDLEERTKNGVIRGQPSPLGAAVINGSQHPSSSSSVNDVSS 401
EmMPK3    VDSQSRSVEEWKEMIFNEVRNFTPRT---------------------------------- 379
SmedJNK   VDERELTVDQWKDLIFRKVKEYESQPENM------------------------------- 378
          :*.:. :::**::*:..:* :   :

HsJNK1    MSTDPTLASDTDSSLEAAAGPLGCCR 427
EmMPK3    -------------------------  379
SmedJNK   -------------------------  378
```

**Fig 5. Sequence features and interactions of EmMPK3.** A) Y2H experiment showing the interaction between EmMKK3 and EmMKK2 with EmMPK3. Left plate, SD-Leu-Trp for transfection control; right plate, SD-Leu-Trp-His with 7,5 mM 3-Amino-1,2,4-triazol. 1, positive control (T-antigen-AD x p53-BD); 2, negative control (T-antigen-AD x lamC-BD); 3, EmMKK2-AD x EmMPK3-BD; 4, EmMKK3-AD x EmMPK3-BD; 5, EmMKK2-AD x empty-BD; 6, EmMKK3-AD x empty-BD; 7, EmMPK3-BD x empty-AD. B) Amino acid sequence alignment of JNK of different origin. Compared are the sequences of *E. multilocularis* EmMPK3 (this work; accession no. MW358638), human JNK1

(HsJNK1; P45983), and *Schmidtea mediterranea* JNK (SmedJNK; AHL18082.1). Sites of perfect alignment (*) as well as groups of strong (:) or weak (.) similarity are marked below the alignment. The JNK-typical TPY motif of the activation loop is shown by a line above the alignment. Residues of the hydrophobic cleft that interact with SP600125 are marked by ‚+' above the alignment.

and EmMKK3 clearly interacted with each other under medium stringency conditions. These data indicated that we had characterized a complete JNK MAPK cascade module of *E. multilocularis* consisting of the MAPKKK EmMEKK1, the MAPKK EmMKK3, and the MAPK EmMPK3. Interestingly, we also found an interaction between EmMPK3 and the previously characterized MAPKK EmMKK2 which, within the *Echinococcus* Erk-branch, is a downstream interaction partner of EmRaf and acts upstream of EmMPK1 [16] (Fig 5).

## Expression of *emmpk3* in *Echinococcus* stem cells

If the encoded proteins of *emmekk1* and *emmpk3* form part of a functional MAPK cascade module, one would expect that they are co-expressed in specific cells of the metacestode. In RT-qPCR experiments on stem cell depleted metacestode vesicles, we could not detect a statistically significant reduction of *emmpk3* transcripts when compared to the control (S2 Fig), indicating that the gene is at least not specifically expressed in GC. To further investigate these aspects, we carried out WISH experiments for *emmpk3* on metacestode vesicles (isolate GH09), combined with EdU labelling of proliferating cells. As shown in Fig 6, *emmpk3* expression could be detected in both, proliferating stem cells as well as in numerous post-mitotic cells in *in vitro* cultivated vesicles of isolate GH09 (Fig 6). On average, around 75.9% of all cells in these vesicles (+/- 5.0%; n = 1074) showed no staining for either EdU or *emmpk3*, 3.2% (+/- 1.8%) stained for EdU only, 16.0% (+/- 6.1%) stained for *emmpk3* only, and an additional 4.9% (+/-1.4%) double-stained for EdU and *emmpk3* (Fig 6). Hence, in total, around 60.4% of all EdU+ cells also stained positive for *emmpk3*, whereas 23.4% of all *emmpk3*+ cells co-stained with EdU. Since 69% of all proliferative stem cells also showed expression of *emmekk1* (see above), these data indicated that at least one third of all proliferative stem cells in metacestode vesicles co-express *emmekk1* and *emmpk3*.

## Effects of inhibitor SP600125 on metacestode vesicles and parasite stem cells

We next investigated the role of EmMPK3 in parasite survival and growth. SP600125 is a widely used, ATP-competitive inhibitor with high specificity for JNK isoforms [45] that has already been used to study the function of JNK in invertebrate model systems such as *Drosophila* [46] and planarians [23,24]. Furthermore, all residues of the JNK hydrophobic cleft that are involved in the interaction between SP600125 and human JNK1 [47] were perfectly conserved in EmMPK3 (Fig 5), indicating that the compound should effectively inhibit the parasite enzyme. We thus performed experiments concerning the effects of SP600125 on parasite metacestode vesicles and stem cells (both from isolate H95). As depicted in Fig 7, SP600125 treated metacestode vesicles lost structural integrity at a concentration of 25 µM after 13 days, and we observed distinct lesion in the germinal layer already after 6 days and even more pronounced after 13 days of incubation. To investigate effects on the parasite stem cell population, we also performed EdU staining experiments on SP600125 treated vesicles. To this end, metacestode vesicles after treatment for 13 days in the presence of SP600125 were thoroughly washed and subsequently incubated for 5 h in EdU. As shown in Fig 7, we observed a concentration dependent decrease of proliferating stem cells which was highly significant after incubation with 25 µM SP600125. In short term exposure experiments (72 h incubation with SP600125), on

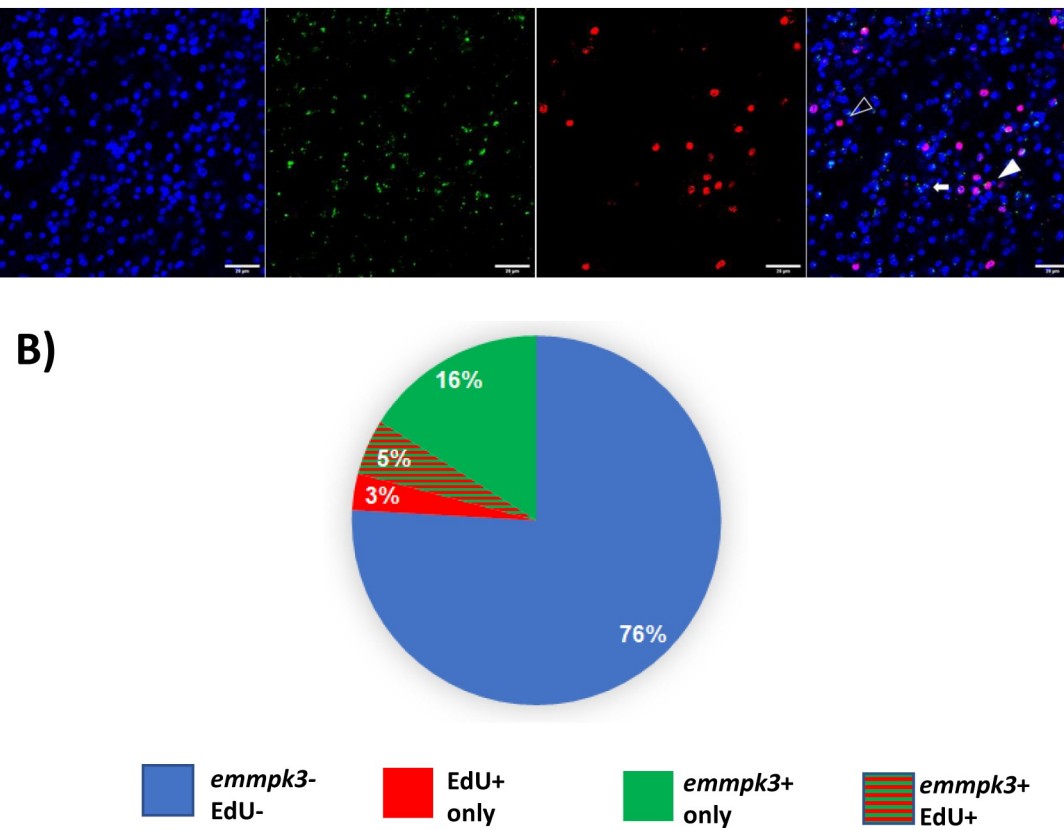

**Fig 6. Expression of *emmpk3* in metacestode vesicles.** A) WISH on vesicle germinal layer. Shown are from left to right: nuclear DAPI staining (blue), *emmpk3* WMISH (green), EdU detection (red), and merge of all channels. Examples cells are marked by full triangle (Edu+, *emmpk3*+), open triangle (Edu+ only) or arrow (*emmpk3*+ only). Bar represents 20 μm. B) Summary of the WISH results. Values are given in percent for cells which are negative for EdU and for *emmpk3* (blue), EdU + cells only (red), *emmpk3*+ cells only (green), and double stained cells EdU+/*emmpk3*+ (red/green).

the other hand, no statistically significant decrease of EdU+ cells was observed, indicating that several cycles of cell division are necessary for a detectable impact on parasite stem cells.

The effects of SP600125 on stem cell proliferation in metacestode vesicles could either result from a direct effect on the germinative cells or be due to a diminished mitotic capacity of the stem cells, caused by damage to the surrounding tissue. We therefore also tested SP600125 on the capacity of primary cell cultures, which are highly enriched (> 80%) in germinative stem cells [2], to form mature metacestode vesicles. As shown in Fig 7, we observed a pronounced decrease of vesicle formation from stem cells which was significant at concentrations of 5 μM SP600125 and above. At 25 μM SP600125, no mature vesicles were formed from primary cells. Taken together, the effects of SP600125 on vesicle integrity, on cell proliferation in metacestode vesicles, and on the capacity of germinative cells to form metacestode vesicles indicated an important role of EmMPK3 in *Echinococcus* stem cell function.

## Discussion

Due to their central role in regulating cellular proliferation and differentiation as well as the fact that they contain exceptionally well druggable enzymes, MAPK cascade modules are

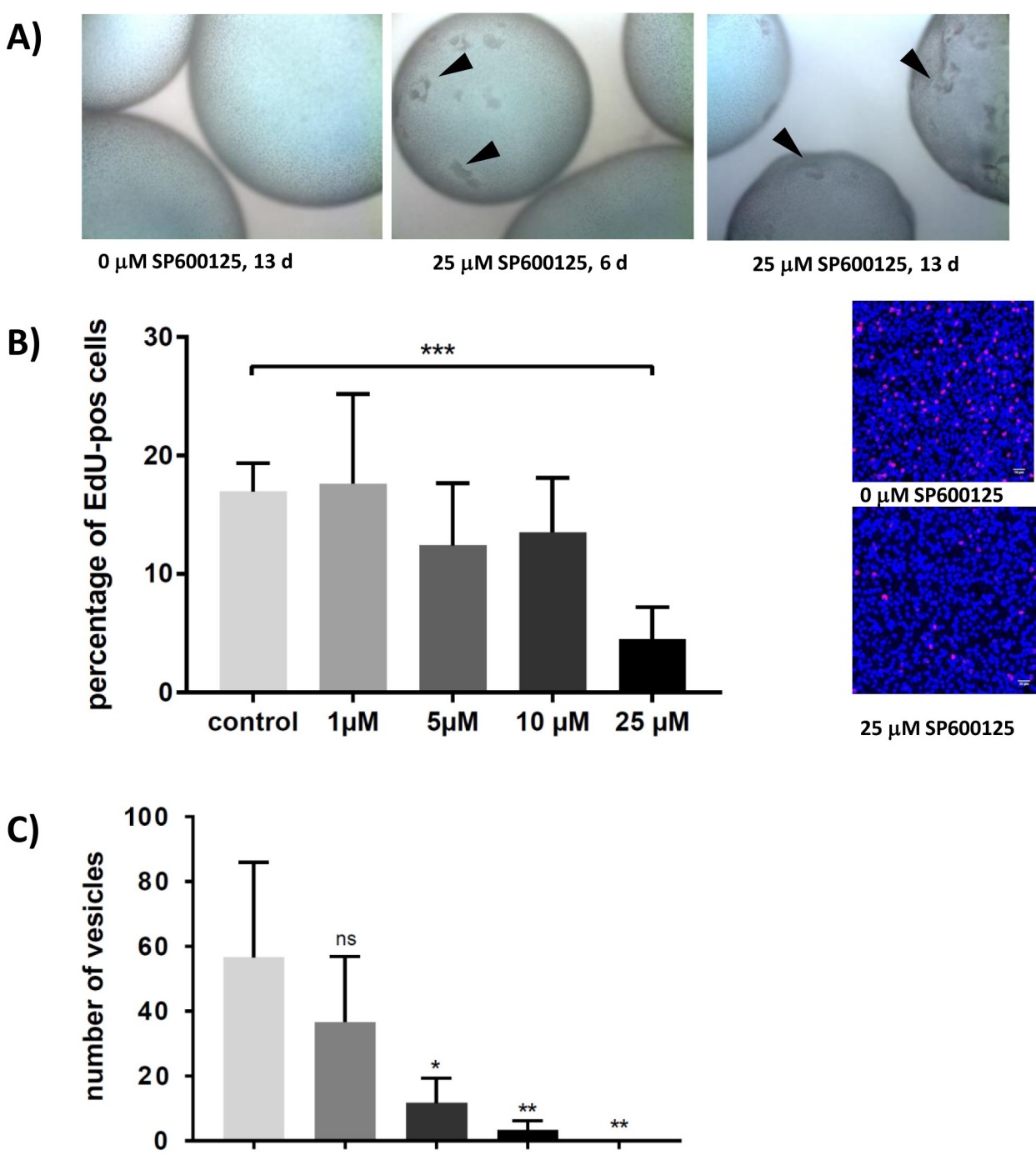

**Fig 7. Effect of SP600125 on metacestode vesicles and parasite stem cells.** A) Effect on metacestode vesicles. Upper panel: Microscopic images of *in vitro* cultivated metacestode vesicles after incubation with 25 μM SP600125. left: control without inhibitor; middle: 25 μM SP600125 for 6 days; right: 25 μM SP600125 for 13 days. Lesions in the germinal layer are marked by black triangles. B) Effect of SP600125 on percentage of EdU + cells in germinal layer. Vesicles had been incubated for 13 days in the presence of SP600125 at concentrations as indicated. Left: dose-dependent effect of SP600125 on number of EdU+ cells in the germinal layer. Statistical analysis of three biological replicates (***, p<0,0001). Right: Examples of EdU+ cells in germinal layer of vesicles incubated with 0 μM (control) and 25 μM SP600125 as indicated. C) Effect of SP600125 on vesicle formation from primary cells. Primary cell cultures had been incubated in the presence of SP600125 at indicated concentrations for 19 days and fully mature vesicles were counted. Statistical analysis of three biological replicates. ns, non significant; *, p = 0,02; **, p<0,008.

attractive targets for the development of chemotherapeutic treatments [13]. In *E. multilocularis* we have previously described several MAPK cascade components such as the MAPKKK EmRaf [14], the MAPKKs EmMKK1 and EmMKK2 [16], and the MAPKs EmMPK1 (Erk-like) [15], and EmMPK2 (p38-like) [17]. At least for one of these components, EmMPK2, we also showed that specific small molecule inhibitors also exhibit profound anti-*E. multilocularis* activity [17]. Likewise, in two recent studies concerning the Erk-like signalling cascade, Cheng et al. [48] and Zhang et al. [49] demonstrated that inhibitors specifically designed against human MEK1 and MEK2 have anti-parasitic activities against the *E. multilocularis* metaces-tode and the *E. granulosus* protoscolex, respectively. In contrast to the Erk-like EmMPK1, which forms a complete MAPK cascade module with EmMKK2 and EmRaf [16], EmMPK2 contains several amino acid changes that render this enzyme constitutively active [17]. Fur-thermore, EmMPK2 does not appear to require activation by interaction with upstream MAPKKs [16,17]. In this work, we now demonstrate that *E. multilocularis* also expresses com-ponents of the third branch of MAPKs, the JNK signalling pathway (Fig 8), which in mammals regulates many crucial processes such as motility, apoptosis, and metabolism [18]. In mam-mals, one of the decisive upstream activators of JNK signalling is the multifunctional MAPKKK MEKK1 the hallmark of which, in addition to the kinase domain, is a RING motif that directs ubiquitination of target proteins [22]. Interestingly, *mekk1* orthologs appear to be absent from well-studied invertebrate model systems such as *Drosophila* and *C. elegans* but at least one such molecule has been described in the free-living flatworm *Dugesia japonica* [26]. We now could clearly show that also the parasitic flatworm *E. multilocularis* expresses a *mekk1* ortholog, *emmekk1*, and that the respective gene is well expressed in the majority of EdU+ (proliferating) stem cells of the parasite. We thus suggest that *emmekk1* fulfils its function either in *Echinococcus* stem cell maintenance or differentiation. Since we also detected *emmekk1* expression in post-mitotic cells, we favour an involvement of *emmekk1* in stem cell differentiation instead of stem cell maintenance or self-renewal. This notion is supported by the general involvement of MAPK cascade modules, particularly the Erk module, in cellular differentiation processes [13]. Furthermore, knockdown of the *mekk1* ortholog in the related *D. japonicum* did not have effects on the number of EdU+ neoblasts but rather led to differen-tiation of neoblasts into anteriorized progeny [26]. Hence, we suggest that *emmekk1* is expressed in *Echinococcus* stem cells during mitosis and directly after mitosis in transitional cells to regulate terminal differentiation processes, although this still has to be experimentally verified.

In planarians, neoblast dynamics and particularly asymmetric cell divisions during stem cell differentiation are regulated by EGF signalling [50]. In *Echinococcus*, EGF signalling through the previously characterized EGF receptor EmER [11] also stimulates stem cell prolif-eration and, thus, most likely influences stem cell differentiation dynamics [10]. One of the crucial downstream effectors of EGF for the transmission of signals in mammals is Grb2, a SH2 and SH3 containing adapter molecule that typically binds to phosphorylated tyrosine resi-dues (consensus: pY-X-N-X) in the C-terminal tail of EGF receptors [21]. Since we observed a strong interaction between EmMEKK1 and the only Grb2-ortholog encoded by the *Echinococ-cus* genome, EmGrb2, in the Y2H system we suggest that in *E. multilocularis* EmMEKK1 is transmitting signals from the EGF receptor EmER through EmGrb2 to downstream MAPKKs. Interestingly, one tyrosine residue within a conserved Grb2 binding motif ($Y_{1447}$YNT) is pres-ent in the C-terminal tail of EmER [11] and could thus serve as a docking site for EmGrb2. Although further experiments are necessary to verify a direct interaction between activated EmER and EmGrb2 we propose that, similar to the situation in mammals, *Echinococcus* uti-lizes Grb2 and MEKK1 orthologs for the transmission of EGF signals to downstream signalling pathways.

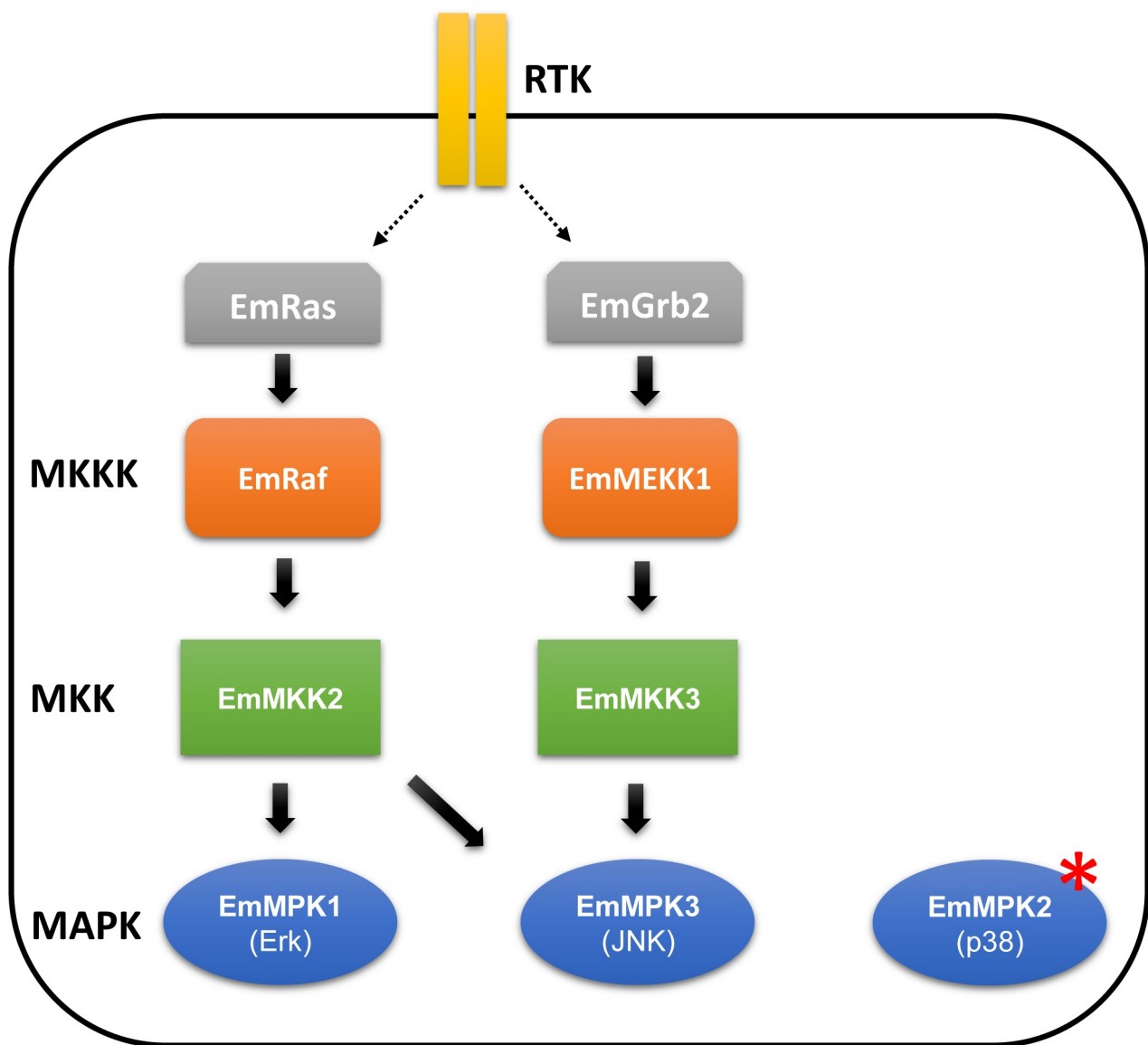

**Fig 8. Model for MAPK cascade interactions in *E. multilocularis*.** Depicted are *E. multilocularis* MAPK, MAPKK (MKK), and MAPKKK (MKKK) as well es upstream regulatory factors identified in this work and previous studies [14–17]. Interactions which are verified by Y2H analyses are shown as solid black arrows, those which are presumed are indicated by dotted black arrows. Red asterix indicates that EmMPK2 is a constitutively active MAPK [17]. RTK, receptor tyrosine kinase.

We cloned and characterized the whole set of *E. multilocularis* MAPKKs and identified by yeast two hybrid analyses one of these, EmMKK3, as a downstream interactor of EmMEKK1. Like in the case of Grb2 this reflects the situation in mammals where MAPKKs of the JNK signalling branch are interactors of MEKK1 [51]. We do not yet know whether EmMEKK1 can also stimulate Erk-signalling in the parasite as is typical for mammals [20] and as has been suggested for planarians [26]. At least EmMKK3 was the only MAPKK that interacted with EmMEKK1 and we could not detect interactions of EmMKK3 with EmMPK1, the only known Erk-like kinase of *Echinococcus* [15]. Besides EmMPK1-3, however, there are still two further MAPKs encoded by the *E. mutilocularis* genome which both contain canonical activation loop motifs for Erk-kinases (T-E-Y). Further experiments are necessary to establish whether

EmMKK3 can interact with one of the molecules which would then create the link between EmMEKK1 and Erk signalling in the parasite.

By Y2H, we observed a clear interaction between EmMKK3 and EmMPK3 which, again, matches the situation in mammals and establishes for the first time the presence of a complete JNK signalling module in *Echinococcus*. Using an inhibitor against mammalian JNK, which has already been used to study JNK signalling in planarians [23,25], we observed detrimental effects on stem cell function in both mature metacestode vesicles as well as in primary cell cultures. In mature metacestode vesicles, SP600125 treatment led to a gradual loss of proliferating stem cells present in the germinative layer, and in primary cell cultures the JNK inhibitor effectively prevented the production of metacestode vesicles from stem cells. We cannot yet tell whether these effects are due to stem cell elimination upon inhibitor treatment, to increased apoptosis associated with JNK signalling, or to cell cycle attenuation. At least in *D. japonica* and *S. mediterranea*, JNK inhibition specifically interfered in G2/M-phase transition [23,24] and it is highly likely that this also led to germinative cell impairment in *Echinococcus*. The phenotype we observed for SP600125 treated vesicles, on the other hand, was different from what we had observed earlier for vesicles in which GC had specifically been eliminated by treatment with hydroxy urea or the Polo-like kinase inhibitor Bi 2536, but which otherwise remained intact for 2–3 weeks of *in vitro* culture [2,36], indicating slow cellular turnover. In our present experiments, we observed structural damage to vesicles earlier. We thus propose that SP600125, in addition to directly affecting *Echinococcus* stem cells, also affects differentiated or differentiating cells, probably those which were *emmpk3*+ but negative for EdU. Of course, we cannot rule out that the elevated concentration of 25 μM of SP600125, at which we observed the most striking phenotypes, also involved off-target effects, particularly since *in vitro* enzyme inhibition assays indicated that SP600125, although being highly selective for JNK over Erk-, and p38-like kinases as well as 14 other kinases initially tested [52], also affects several other mammalian kinases with similar $IC_{50}$ [53]. In planaria, sub-lethal doses of SP600125 (up to 5 μM) phenocopied the tailless phenotype induced by JNK-specific RNAi [25], indicating that at least at lower doses, SP600125 appears to have little off-target effects in flatworms. At concentrations of 25 μM, SP600125 was lethal to *S. mediterranea* within very short time (30 min to 1 h of exposure) [25]. Since in our experiments, metacestode viability was observed at least for 7 d, we suppose that we also had applied sub-lethal doses of SP600125 (maybe quenched by the large amount of hydatid fluid in the vesicles) and that at least the majority of observed phenotypes, particularly the inhibition of primary cell cultures at 5 μM, was due to inhibition of EmMPK3. As previously reported by Chu et al. [52], the likeliest alternative target of SP600125 in humans is the mitotic checkpoint kinase Mps1 (also known as TTK) and, interestingly, the *E. multilocularis* genome also encodes a member of the Mps kinase family (EmuJ_000595100) which is expressed in primary cell cultures and metacestode vesicles [27] (S3 Fig). By structural analyses, Chu et al. [54] had identified 8 amino acid residues of particular importance for the interaction between SP600125 and the ATP-binding pocket of Mps1. Since in the *Echinococcus* Mps kinase, 4 of these residues are identical and the remaining residues possess similar biochemical features (S3 Fig), off-target effects of SP600125 on the EmuJ_000595100 gene product cannot be excluded. Further biochemical analyses are thus necessary to distinguish between specific effects of SP600125 on EmMPK3 and the possibility of off-target effects on the *Echinococcus* Mps kinase.

One of the most striking features of the *Echinococcus* metacestode is that is consists of posteriorized tissue in which anterior marker genes like *sfrp* are completely absent whereas posterior markers such as *wnt1* are expressed throughout the germinative tissue [5]. It is thus highly likely that posterior ligands of the *wnt*-pathway, which are produced by muscle cells within the germinative layer [5], are crucial instructors of *Echinococcus* GC to produce posteriorized

progeny. Interestingly, apart from being involved in cell cycle regulation of neoblasts in free-living flatworms [23,24], JNK signalling also appears to determine posterior fate of differentiating stem cells. As demonstrated by Tejada-Romero et al. [25], pharmacological inhibition of planarian JNK (using SP600125) as well as RNAi against JNK led to tailless phenotypes and to prevention of *wnt*-dependent cell responses necessary for establishing the posterior pole. This also might involve the planarian *mekk1* ortholog since RNAi against *D. japonica* MEKK1 led to substantial defects in anterior-posterior patterning [26] although the direct link between planarian MEKK1 and JNK has not been established so far. The effect of pharmacological inhibition of EmMPK3 on primary cell cultures which effectively prevented the formation of metacestode vesicles from stem cells could, thus, also be due to substantial interference with the parasite's *wnt* signalling pathway. At least in mammalian systems and in planaria, substantial cross-talk between components of JNK- and *wnt*-signalling has been previously observed [25]. The possibility of such interactions is currently subject to investigations in our laboratory.

In conclusion, in this work we characterized for the first time a JNK signalling module of *E. multilocularis* that is composed of a MEKK1-like MAPKKK, EmMEKK1, a MAPKK of the JNK branch, EmMKK3, and a JNK-like molecule, EmMPK3. We demonstrate that this module is expressed in *Echinococcus* stem cells and that inhibition of EmMPK3 by JNK inhibitors leads to loss of proliferating stem cells in mature metacestode vesicles as well as to developmental defects that prevent the formation of metacestode vesicles from parasite stem cells. We thus propose that, similar to the situation in mammals and planaria, JNK signalling fulfils an important role in stem cell maintenance and dynamics. Furthermore, due to well established cross-talk between JNK- and *wnt*-signalling pathways, EmMEKK1 and EmMPK3 might be important signal transducers which lead to posteriorized development as it is characteristic for the *Echinococcus* metacestode. Due to the apparent importance of the JNK signalling pathway for *Echinococcus* stem cell function and to the amenability of its components for chemical inhibition, this opens new ways for the development of anti-echinococcosis drugs. To this end, structural differences in the ATP binding pockets of EmMPK3 and mammalian JNK, similar to those between human JNK1-3 and the Mps1 kinase [54], would have to be exploited for structure-based drug design to generate compounds that are highly selective for the parasite enzyme.

## Supporting information

**S1 Table. List of genes investigated in this study and primer sequences.**
(XLSX)

**S1 Fig. Expression of *E. multilocularis* genes in larval stages.**
(PDF)

**S2 Fig. RT-qPCR analyses for the expression of *emmekk1* and *emmpk3* in stem cell-depleted metacestode vesicles.**
(PDF)

**S3 Fig. Amino acid sequence alignment of human and *Echinococcus* Mps kinases.**
(PDF)

## Acknowledgments

The authors are indebted to Raphael Duvoisin and Stefanie Riedl for providing vectors constructs and wish to thank Dirk Radloff for excellent technical assistance.

## Author Contributions

**Conceptualization:** Klaus Brehm.

**Data curation:** Kristin Stoll, Monika Bergmann, Markus Spiliotis, Klaus Brehm.

**Formal analysis:** Kristin Stoll, Monika Bergmann, Markus Spiliotis, Klaus Brehm.

**Funding acquisition:** Klaus Brehm.

**Investigation:** Kristin Stoll, Monika Bergmann, Markus Spiliotis, Klaus Brehm.

**Methodology:** Kristin Stoll, Monika Bergmann, Markus Spiliotis, Klaus Brehm.

**Project administration:** Klaus Brehm.

**Supervision:** Markus Spiliotis, Klaus Brehm.

**Validation:** Kristin Stoll, Monika Bergmann, Markus Spiliotis, Klaus Brehm.

**Visualization:** Kristin Stoll, Monika Bergmann, Markus Spiliotis, Klaus Brehm.

**Writing – original draft:** Kristin Stoll, Monika Bergmann, Markus Spiliotis, Klaus Brehm.

**Writing – review & editing:** Klaus Brehm.

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
