## [Decision Letter · Decision Letter 0]

30 Sep 2021

Dear Prof. Brehm,

Thank you very much for submitting your manuscript "A MEKK1 – JNK mitogen activated kinase (MAPK) cascade module is active in Echinococcus multilocularis stem cells" for consideration at PLOS Neglected Tropical Diseases. As with all papers reviewed by the journal, your manuscript was reviewed by members of the editorial board and by several independent reviewers. The reviewers appreciated the attention to an important topic. Based on the reviews, we are likely to accept this manuscript for publication, providing that you modify the manuscript according to the review recommendations. 

Please address the suggestions made by reviewers. Thankyou

Sincerely,

Michael J. Smout

Associate Editor

Cinzia Cantacessi

Deputy Editor

Please address the suggestions made by reviewers. Thankyou

Reviewer's Responses to Questions

**Key Review Criteria Required for Acceptance?**

**Methods**

-Are the objectives of the study clearly articulated with a clear testable hypothesis stated?

-Is the study design appropriate to address the stated objectives?

-Is the population clearly described and appropriate for the hypothesis being tested?

-Is the sample size sufficient to ensure adequate power to address the hypothesis being tested?

-Were correct statistical analysis used to support conclusions?

-Are there concerns about ethical or regulatory requirements being met?

Reviewer #1: The paper of Stoll et al. deals with molecular processes underlying the development of metacestode larvae of E. multilocularis that grow in a tumour-like manner within host tissues such as the liver. In this comprehensive and elegant study, the authors provide first evidence that a specific MAPK cascade is important in stem cells but also in differentiated cells of this parasite. Additional studies with a JNK inhibitor showed reduced metacestode vesicles and a reduction of proliferating stem cells in mature metacestode vesicles. This justifies speculations about the potential of JNK inhibitors as novel chemotherapeutics against echinococcosis.

Reviewer #2: (No Response)

Reviewer #3: (No Response)

Reviewer #4: this is a classical hypothesis driven research report, with clear objectives and experiments carried out with great expertise. The only question I have relates to the Echinococcus multilocularis isolates that are used. The authors indicate that they used E. multilocularis isolates(H95, GH09, Ingrid, J2012. . Are there differences between these isolates in terms of growth characteristics, protoscoleces formation or any other features such as virulence in mice?

**Results**

-Does the analysis presented match the analysis plan?

-Are the results clearly and completely presented?

-Are the figures (Tables, Images) of sufficient quality for clarity?

Reviewer #1: -yes

-yes

-yes

Reviewer #2: (No Response)

Reviewer #3: (No Response)

Reviewer #4: The analysis was done with great expertise, and figures clearly present the results. The authors show very nicely how the different components of this module interact with each other, and they also show that treatment with a JNK inhibitor led to reduced formation of metactodes from stem cells, as well as a specific reduction of stem cells in mature metacestodes. They indicate that JNK inhibitors could be used as novel chemotherapeutics against echinococcosis.

**Conclusions**

-Are the conclusions supported by the data presented?

-Are the limitations of analysis clearly described?

-Do the authors discuss how these data can be helpful to advance our understanding of the topic under study?

-Is public health relevance addressed?

Reviewer #1: -yes

-yes

-yes

-yes

Reviewer #2: (No Response)

Reviewer #3: (No Response)

Reviewer #4: The paper is very well written, experiments were done with great expertise, results are presented in a concise and clear manner, and most of the conclusions drawn from the results are sound. A few points to consider, especially in relation to the inhibitor studies:

SP600125: how specific is this inhibitor? The manuscript indicates that this inhibitor acts on metacestodes and germinative cells in the µM-range, while in mammalian cells this inhibitor works well at nM concentrations. What would be the normal concentration used for this inhibitor? Is there a risk that the massive amounts used to inhibit Echinococcus could produce off-target effects?

In relation to this comment, ideally the authors should show inhibitor studies carried out directly on EmMPK3 activity, not only on stem cells or whole metacestodes. 

- Based on the efficacy of the inhibitor, I strongly believe that the statement that JNK inhibitors could be candidates for chemotherapeutical treatment is not really supported by the data.

**Editorial and Data Presentation Modifications?**

Reviewer #1: Major comment

In their EdU analyses, the authors observed a difference between the isolates Ingrid and J2012 (text and Fig 2B). Can the authors speculate about a presumptive reason of this difference? Could this have something to do with the origin of these isolates, is there a different genetic equipment? 

Minor comments

line 55: In this work, … 

line 58: … in parasite stem cells, which drive …

line 74: … infectious eggs, which are released by the definitive host and are orally taken up by the intermediate. (Here and elsewhere, insert commas in front of “which”)

line 94: During recent years, we have …

line 156: Insert space between …J20212) and [29]…

line 161: define (d) for days and use abbreviation afterwards.

line 188: define WMISH here (to my knowledge, whole mount in situ hybridization is abbreviated as WISH) and use only the abbreviation later on (see e.g. Fig. 6 legend)

line 193 following: hours (h). Use abbreviation afterwards.

line 309: delete “could”; better: … we detected …

line 350: delete the second “been” 

line 359: delete “well” 

lines 361/367: axins

line 377: In the legend of Fig 3, Y2H has been defined as abbreviation. This should be done in the text upon first use of the term yeast two hybrid. 

line 401: delete “could”; better: … we identified …

line 409: Echinococcus should be given in italics (please check italics writing for species names throughout the manuscript; see e.g. line 517: Drosophila)

line 514: We next investigated … 

line 515: ATP-competitive

line 542: italics writing for „in vitro“ should be harmonised throughout the manuscript

line 577: In this work, … 

line 629: delete “could”; better: By Y2H, we observed …

References: mixture of upper and lower case use for paper titles, please harmonise.

Reviewer #2: (No Response)

Reviewer #3: (No Response)

Reviewer #4: No comments

**Summary and General Comments**

Reviewer #1: The manuscript is well written, and its content perfectly fits to the aims and scope of PLoS NTD. As such, it will be of high interest for the community. I have only little to critisize.

Reviewer #2: This paper presents another important advance from this lab in the understanding of signal transduction and growth in tapeworms. It is well written and clearly presented (albeit so littered with acronyms in places it can become almost non-prose) and the methods and interpretations are sound. I would like to have seen the genome-wide effects of the inhibition assays via transcriptome profiling, but there is more than enough functional and other evidence to support their stated conclusions and hypotheses. I recommended it highly for publication.

Minor comments and corrections:

81. This is a characteristic of all flatworms – why not state that, unless to leave the impression it’s unique to E. mul. or some other exclusive group of flatworms?

166. I was immediately wondering why there was no discussion of how were GOIs chosen/identified – however, reading further I found this was discussed in detail in multiple places. Perhaps here you could mentioned that fact – eg. “Identification of GOIs is discussed in conjunction with results below.”

266 WormBase

325. italicise emmekk1

331. This final conclusion should be joined to the end of the preceding paragraph.

390. Add parentheses around “*” and “:”

409. Italicise Echinococcus

517. Italicise Drosophila

556. As above, this final conclusion should be joined to the end of the preceding paragraph (only paragraphs are offset from other paragraphs – and by definition a paragraph consists of > one sentence)

608. A figure of the proposed signalling cascade would be helpful

643. Italicise Echinococcus

663. Transcriptome profiling of these assays would have been incredibly informative for tracking the full downstream affects of the inhibition.

Happy birthday KB!

Reviewer #3: This manuscript describes the characterization of MEKK1/JNK signaling cascade in E. multilocularis, the last module of the MAPK cascades that has not been identified yet in the parasite. The authors identified key members of the JNK signaling pathway and showed that the JNK inhibitor SP600125 inhibits the formation of metacestode vesicles from stem cells and reduces the number of proliferating cells in vesicles, suggestive a role of the JNK signaling in regulating E. multilocularis stem cells. The manuscript is well presented and well written. Overall, I endorse this work but do have a few concerns and suggestions.

Major concerns:

1) The statements that the authors addressed in Abstract “that are most likely germinative cell progeny” and L331-333 “these analyses indicated that emmekk1 is expressed …”. 

I’m afraid that there is lack of strong evidence to conclude that the emmekk1+EdU- (EmMPK3+EdU-) cells are immediate stem cell progeny. I think the best way to obtain direct evidence is using a sequential EdU/BrdU pulse staining followed by WMISH, which is, however, technically difficult in the parasite so far. Alternatively, the authors, to make this issue clearer, may check the emmekk1+ (EmMPK3+) cells in the vesicles after a long period of time of EdU incubation (e.g. 24 h) or after different time of EdU incubation (e.g. 5-24 h), or after a treatment with hydroxyurea which eliminates most germinative cells. Otherwise, I suggest the authors should soften this conclusion accordingly in the results and discussion. 

2) L374-375: “indicated that EmMEKK1 could act as a signal transducer of EGF signaling in E. multilocularis via interaction with EmGrb2” 

These results only indicate that EmMEKK1 interacts with EmGrb2, however, whether the interaction responds to EGF signal remains largely uncertain in the parasite. I suggest the authors tone down their statement on this. 

Others:

1) The authors found that the vesicles lost their structural integrity after 6-13 days of SP600125 treatment (Fig.7). This is an interesting result. It was shown that after specific elimination of stem cells by hydroxyurea the vesicles usually remain structurally intact for at least two weeks. A similar result was also observed by the same group when using the PLK1 inhibitor BI2536, which targets specifically the mitotic cells. Taken together, these results suggest that inhibition of the JNK signaling by SP600125 may not only affect the stem cells, and/or that SP600125 may have targets other than JNK in the parasite. I suggest the authors could make some discussion on this. 

2) Fig.7. The title for y-axis in Fig.7B. Is it the percentage of EdU+ cells of total cells in the vesicles? The ratio for ctrl is > 15%, much higher than those as shown in Fig.2 and 6 (~8%). In addition, did the authors check the number of EdU+ cells and the expression of some proliferation markers (e.g. h2b) in the vesicles after a short term of treatment with SP600125 (24-72 h)? According to the results presented here, it seems that, in spite of the high expression of EmMPK3 in ~2/3 of the proliferating stem cells, SP600125 impairs the maintenance rather than the proliferation of the stem cells. 

3) The authors quoted the transcriptome data (ref. 27) several times to indicate the gene expression. However, I think it is better to perform q-PCR experiments to verify this, especially for those genes of which expression is high in the stem cell populations. 

4) editorial suggestions

L80: the exclusively-> exclusively

L263: highest-> the highest

L271: 1.462 amino acids -> 1,462 amino acids

L437: any of the EmMEKKs -> any of the EmMKKs?

L438: with EmMEKK3 -> with EmMKK3?

L480: interacted with each other

L495-499: 75,9% ->75.9%; 5,0%->5.0% ...

Fig. 6B: I suggest using ‘emmpk3-EdU-’ instead of ‘DAPI only’

Reviewer #4: Overall, this study presents novel information on potential signaling pathways in Echinococcus. However, the statement that this study shows that JNK pathways represent novel targets for chemotherapy is not supported by the presented data. I think this statement could be easily removed without impairing the overall impact and importance of this paper.

PLOS authors have the option to publish the peer review history of their article (what does this mean?). If published, this will include your full peer review and any attached files.

Reviewer #1: No

Reviewer #2: Yes: Peter D. Olson

Reviewer #3: No

Reviewer #4: No

Figure Files:

Data Requirements:

Reproducibility:

References

---

## [Editor Report · Decision Letter 1]

25 Nov 2021

Dear Prof. Brehm,

We are pleased to inform you that your manuscript 'A MEKK1 – JNK mitogen activated kinase (MAPK) cascade module is active in Echinococcus multilocularis stem cells' has been provisionally accepted for publication in PLOS Neglected Tropical Diseases.

Best regards,

Michael J. Smout

Associate Editor

Cinzia Cantacessi

Deputy Editor

none

---

## [Editor Report · Acceptance letter]

3 Dec 2021

Dear Prof. Brehm,

We are delighted to inform you that your manuscript, "A MEKK1 – JNK mitogen activated kinase (MAPK) cascade module is active in Echinococcus multilocularis stem cells," has been formally accepted for publication in PLOS Neglected Tropical Diseases.

Best regards,

Shaden Kamhawi

co-Editor-in-Chief

Paul Brindley

co-Editor-in-Chief
